# No effect of multi-axis dot pattern symmetry on subjective duration

**Alexis David James Makin** *, **Afzal Rahman**, **Marco Bertamini**

Department of Psychological Sciences, University of Liverpool, Liverpool, United Kingdom

* alexis.makin@liverpool.ac.uk

## Abstract

Previous work has shown that symmetrical stimuli are judged as lasting longer than asymmetrical ones, even when actual duration is matched. This effect has been replicated with different methods and stimuli types. We aimed to a) replicate the effect of symmetry on subjective duration, and b) assess whether it was further modulated by the number of symmetrical axes. There was no evidence for either effect. This null result cannot be explained by reduced statistical power or enhanced floor or ceiling effects. There is no obvious stimulus-based explanation either. However, we are mindful of the reproducibility crisis and file drawer problems in psychology. Other symmetry and time perception researchers should be aware of this null result. One possibility is that the effect of symmetry on subjective duration is limited to very specific experimental paradigms.

**Data Availability Statement:** Results are available on Open Science Framework, along with the PsychoPy code for running the experiment (https://osf.io/a3u6e/). Other researchers can use this material freely.

## Introduction

### The scientific value of null results

Many leading researchers doubt general trustworthiness of peer-reviewed science [1, 2]. Psychology faces acute criticism because scientific malpractice is allegedly routine, and because there have been famous attempts to estimate reproducibility. Approximately just 39% of published results in psychology are reproducible, and this only increases 51% in cognitive psychology [3]. One factor contributing to the replication crisis is well understood: Scientists publish or perish, and the only readily publishable narrative has the standard happy ending: 'the expected effect was statistically significant (p < 0.05)'. Consequently, the proverbial file drawer accumulates null results, and the literature accumulates false positives (alongside the real effects). Even without blatant data fraud, p-hacking or HARKing (Hypothesising After Results Known), the published record is likely biased towards 'lucky' experiments that overestimate true effect size [4]. Such systemic problems have been obvious for decades, but conveniently ignored.

Keith Laws [5] provides one useful and provocative commentary. He quotes Francis [6]:

*"The scientific method is supposed to be able to reveal truths about the world, and the reliability of empirical findings is supposed to be the final arbiter of science; but this method does not seem to work in experimental psychology as it is currently practiced."*

*(page 3)*

**Funding:** This work was funded by the Economic and Social Research Council UK award to AM https://esrc.ukri.org/ ES/S014691/1 The funders had no role in study design, data collection and analysis, decision to publish, or preparation of the manuscript.

**Competing interests:** The authors have declared that no competing interests exist.

While this particular comment falls on pessimistic end of the spectrum, even the most optimistic commentators acknowledge substantial room for improvement. Laws [5] encourages practicing scientists to show leadership and embrace the challenges. However:

*"This leadership will however require psychologists to take a more active role in submitting replications and null findings—science is clearly not self-correcting."*

*(page 7)*

Laws [5] further recommends that 'failed' experiments should not be marginalized in specialist null results journals:

*"Although laudable, such journals create a special space for replications and null findings rather than acknowledging their place in the centre of science."*

*(page 7)*

Following this advice, we report our failure to replicate an effect from two of our previous papers [7, 8].

## The published effect: Symmetry reportedly elongates subjective duration

Ogden et al. [7] found that symmetrical patterns were misreported as staying on the screen longer than random patterns. In Experiment 1 of Ogden et al. [7], participants were presented with 10 X 10 checkerboard patterns for durations of 500, 750, 1000, 1250 or 1500 ms. The checkerboards had 40 black and 60 white squares. The arrangements were either two-fold reflectional symmetry, 90-degree rotational symmetry, or random. Participants estimated duration and entered their estimates with a computer keyboard. Reflection was judged to have lasted longer than rotation or random (main effect of Regularity, $F_{(2,42)} = 17.114$, $p < 0.001$, $\eta_p^2 = 0.451$). This was replicated in Experiment 2 with an alternative temporal bisection procedure and non-parametric statistical tests. In an earlier study with a more general research question, Palumbo et al. [8] also found that symmetrical patterns were judged longer than random patterns ($F_{(1, 23)} = 5.532$, $p = 0.028$, $\eta_p^2 = .194$). This was mistyped as ($F_{(1,23)} = 5.332$, $p = 0.028$) in Palumbo et al. [8]. We re-ran the analysis. Sasaki and Yamada [9] subsequently reported a comparable 'regular is longer' effect, although their grid-regularity is arguably a different visual category to our reflectional symmetry.

## An experiment on number of folds

Research on symmetry and duration can only progress beyond these initial observations if we establish some basic parameters of the effect. For instance, does the effect of symmetry on duration scale with the salience of symmetry, or is it an all-or-nothing phenomenon? The most reliable way to manipulate symmetry salience is to vary the number of folds, we began with this.

If established, this could be a useful clue for understanding how duration estimation works at a neural level, which remains uncertain despite decades of research [10–12]. Duration estimates could be mediated by an internal pacemaker-accumulator clock [13, 14]. However, the internal clock model is controversial [15, 16] and substantially different alternatives have been proposed [10]. New empirical evidence from exploratory studies like ours could help constrain future theories of time perception.

Example stimuli with 1–5 folds are shown in Fig 1. These particular dot patterns were chosen because they were used in Event Related Potential (ERP) research by Makin et al. [17] and

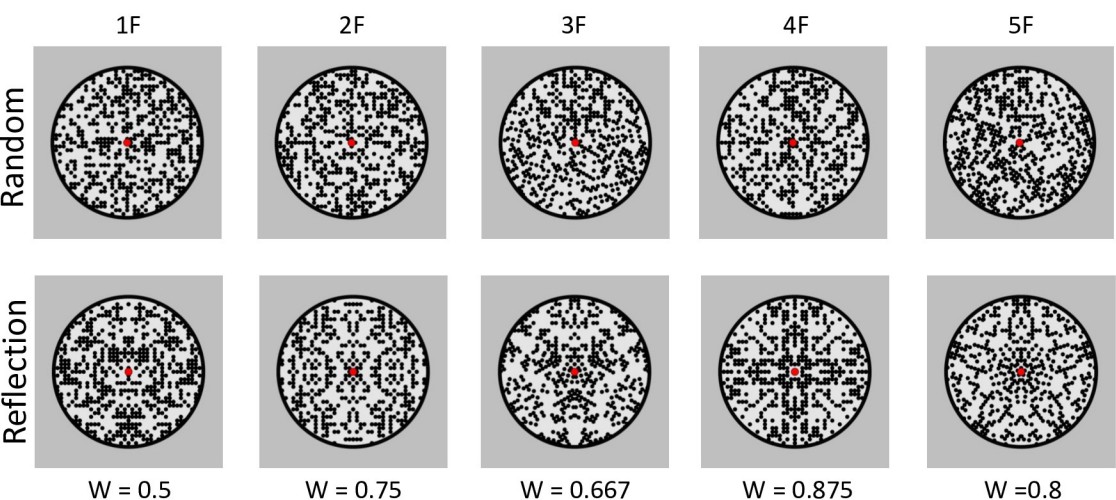

**Fig 1. Example stimuli and W scores from the holographic model.**

again in preference research by Makin et al. [18]. Using the same stimuli across research programmes can sometimes facilitate discovery of new connexions which would otherwise be missed. For instance, there could be an intriguing link between the amplitude of the neural response and subjective duration, suggesting time is coded as sensory excitation. By using consistent stimuli, such links are more likely to be revealed.

Such dot patterns are also valuable because their salience (or 'perceptual goodness') can be quantified with *holographic weight of evidence model* [19, 20]. The holographic model assigns a 'W-load' to dot patterns. This is determined with the $W = E/N$ formula, where W is perceptual goodness, E is the number of 'holographic identities' in a pattern, and N is the number of dots. A 'holographic identity' is a regular substructure with the same regularity as the global regularity in which it is embedded (e.g. a reflected pair of dots in a global reflection). As explained by van der Helm [21], there is then a non-monotonic relationship between W and the number of folds (1F W = 0.5, 2F W = 0.75, 3F W = 0.667, 4F W = 0.875, 5FW = 0.8). Makin et al. [17] found that W predicted the amplitude of neural symmetry response successfully (the expected dip at 3F and 5F was evident in the at around 400 ms). We therefore predicted that the subjective elongation would also scale with W (as determined by the number of folds).

## Methods

### Participants

There were 40 participants (age = 18–55 years, 9 men, 1 left-handed). All had normal or corrected-to-normal vision and gave written informed consent. The experiment was approved by the Ethics Committee of the University of Liverpool and were conducted in accordance with the Declaration of Helsinki (2008).

### Power analysis

Power analysis is a recurring topic in the meta-science literature. Bishop [1] lists low statistical power as one of the 'four horsemen of irreproducibility'. Brysbaert [4] suggests that psychologists routinely misunderstand power analysis, and many research programs are built on an unstable foundation of underpowered experiments (see also Lakens, [22]). A fundamental

problem is that meaningful power analysis requires a reliable estimate of effect size. Brysbaert [4] claims that '*you need an estimate of effect size to get started, and it is very difficult to get a useful estimate" (page 7)*. We have just two relevant a priori estimates of effect size at our disposal ($\eta_p^2 = 0.194$ from Palumbo et al. [8] and $\eta_p^2 = 0.451$ from Ogden et al. [7]). Converting to Cohen's $d_z$ gives us 0.491 and 0.906. A conservative approach is to use the smaller of the two for power analysis.

With this smaller effect size of $\eta_p^2 = 0.194$ ($d_z = 0.491$) observed power was just 0.616. If this is the true population effect size, we have an unacceptably high chance of missing the effect by using another sample of 24 (p = 1–0.616 = 0.384). However, with 40 participants, we have an 0.850 chance of finding the effect. The required sample size for finding a modulatory effect of folds is unclear because we do not have an a priori estimate of effect size here.

## Apparatus and stimuli

Participants sat in a darkened cubicle 140 cm from a 29 X 53 cm LCD monitor. The experiment stimulus was controlled using PsychoPy software [23]. Participants entered their responses on keyboard when prompted. A different pattern was generated on every trial, so no participant ever saw the same exemplar twice.

The stimulus construction algorithm is described in Makin et al. [17] and Palumbo et al. [24]. The circular frame was approximately 5 degrees in diameter, as in Makin et al. [17]. The patterns were generated by filling an implicit grid within pie-slice segments of the circle. The folds produced seams between the segments and local orientation cues, so it was essential to included 1–5 folds in the random condition as well, although these are not reflectional axes in any sense (Fig 1 top row). Each cell could be occupied with a black dot, and the probability of occupation was set as at 0.4. This meant 40% of cells were occupied with a dot. The average number of dots (N-dots) was 525, although this mean varied with folds [1F = 514, 2F = 528, 3F = 521, 4F = 531, 5F = 532]. More importantly, there was some variability around the mean. The SD of N-Dots changed with number of folds and was higher for reflection [1F = 24, 2F = 37, 3F = 44, 4F = 57, 5F = 49] than random [1F = 14, 2F = 15, 3F = 17, 4F = 21, 5F = 17]). Although N-dots (and thus luminance) was more *variable* in the symmetry conditions, *average* N-dots (and thus *average* luminance) was the same in symmetry and random conditions.

## Procedure

The experiment used a 2 x 5 x 3 within-subjects design. The factors were Regularity (reflection and random), Number of folds (1, 2, 3, 4 or 5) and Presentation duration (0.4, 0.6 and 0.8 seconds). There were 10 repeats of each *experimental trial* condition, and thus 300 experimental trials in total.

There were also an additional 300 *filler trials*. The filler trials were also fixed in terms of regularity (reflection, random) and folds (1–5). Filler trial duration was selected from a uniform distribution between 0.2 to 1.0 seconds. Average filler trial duration was thus 0.6, as in the experimental trials. The filler trials prevented participants from overlearning the three durations of the experimental trials.

At the end of each trial, participants used the mouse to rate duration estimates from 0 seconds to 1 second. The scale had 0.01 second increments, so it was effectively continuous (unlike a Likert scale with 7 points).

Trials were presented in a randomized order within blocks of 60 trials. Every trial type was presented once within a block. There was a break in the experiment every 30 trials, when the participant was instructed to press space if they wished to continue.

### Analysis

Most analysis was conducted on the experimental trials. For each participant, the average duration estimates were obtained across the 10 trials in 30 conditions. These were submitted to 2 Regularity X 3 Duration X 5 Folds repeated measures ANOVA. All 30 variables were normally distributed according to Shapiro-Wilk test ($p > 0.07$).

We conducted exploratory analysis of the filler trials. These were submitted to 2 Regularity X 5 Folds repeated measures ANOVA. Again all 10 variables were normally distributed according to the Shapiro-Wilk test ($p > 0.614$). The Greenhouse-Geisser correction factor was employed when the assumption of sphericity was violated (Mauchly's test $p < 0.05$). Adjusted degrees of freedom are reported.

On filler trials, we also measure slope and intercept of the relationship between objective and subjective duration, as well as the correlation between objective and subjective duration. Although each participant was presented with a slightly different distribution of filler trial durations, these three metrics would be identical for ideal observers who always make veridical duration estimates (intercept 0, slope 1, $r = 1$). We can thus use these metrics to assess whether performance on filler trials is influenced by regularity.

### Open science policy

Results are available on Open Science Framework, along with the PsychoPy code for running the experiment (https://osf.io/a3u6e/). Other researchers can use this material freely.

## Results

Duration estimates from the experimental trials are shown in Fig 2A. Although participants typically underestimated duration, subjective duration increased alongside objective duration ($F (1.129, 44.021) = 217.481$, $p < 0.001$, $\eta_p^2 = 0.848$, linear contrast $F (1,39) = 230.530$, $p < 0.001$, $\eta_p^2 = 0.855$). There was no effect of Regularity ($F < 1$) and no effect of Folds ($F < 1$). There was no interaction between Regularity and Duration ($F (2,78) = 1.835$, $p = 0.167$) and no other interactions ($F < 1$).

Next, we analysed the filler trials. As can be seen in Fig 2B, there were no effects of Regularity ($F < 1$) or Folds ($F (4,156) = 1.210$, $p = 0.309$) and no Regularity X Folds interaction ($F (4,156) = 1.409$, $p = 0.233$). Durations were again underestimated in filler trials, (mean estimate 0.461 seconds instead of the veridical 0.6 seconds, Fig 2B).

In the filler trials, most participants showed a strong correlation between objective and subjective duration. This shows that they were sensitive to filler trial duration. Mean correlation coefficients was very similar whether patterns were reflection or random (Fig 2C). This difference was not significant ($t (39) = 0.116$, $p = 0.908$). Finally, we calculated intercepts and slopes of this relationship. These metrics were also unaffected by Regularity (Fig 2D and 2E, $t (39) = -0.479$, $p = 0.634$; $t (39) = 0.674$, $p = 0.505$).

## Discussion

Despite our predictions, there was no evidence that reflectional symmetry dot patterns were judged as lasting longer than random dot patterns, and no evidence that this effect was further modulated by the number of folds. The effect of symmetry on subjective duration was reported by Palumbo et al. [8] and Ogden et al. [7], but we are now less confident that this effect is robust. However, leaving it the current null result unpublished would be to contribute to the file drawer problem, and all perversions of science that follow [4, 5].

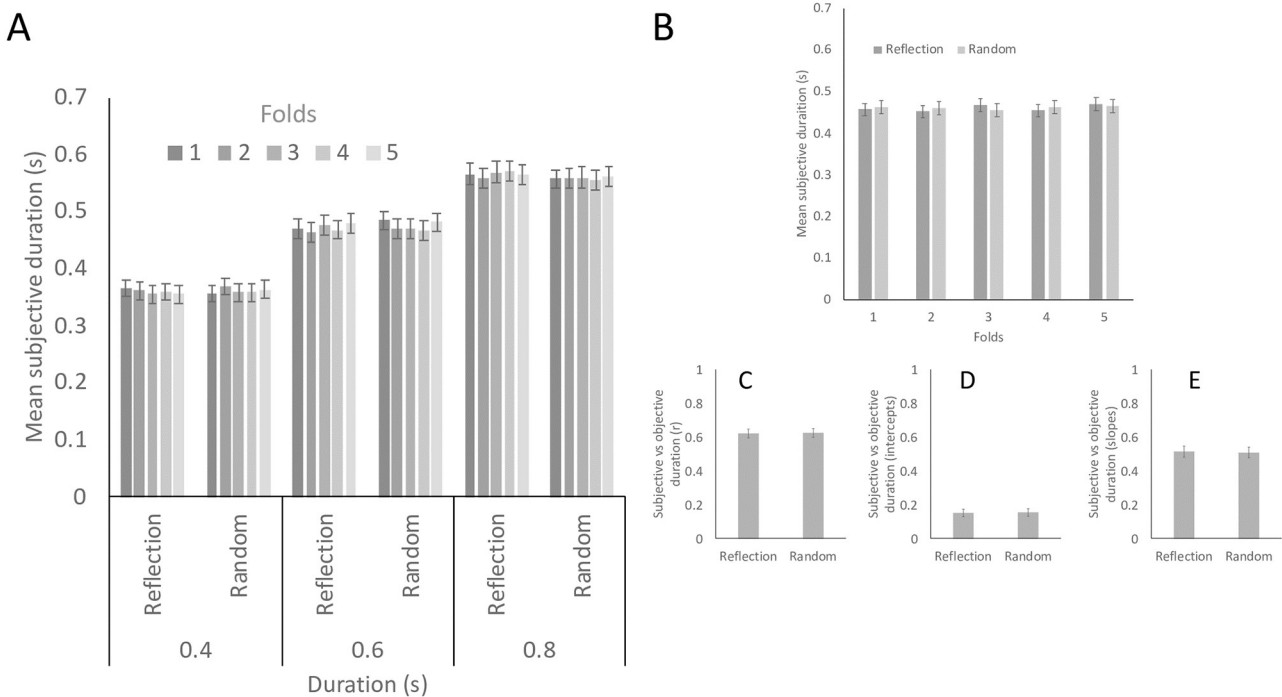

**Fig 2. Results.** A) Mean subjective duration in experimental trials. This is separated into reflection and random conditions with 1–5 folds. B) Mean subjective duration in filler trials in reflection and random conditions with 1–5 folds. C) Mean correlation of the relationship between objective and subjective duration in the filler trials. D) Intercepts of the relationship between objective and subjective duration in the filler trials. E) Slopes of the relationship between objective and subjective duration in the filler trials. Error bars = +/- 1 S.E.M.

An effect of regularity on subjective duration was also reported by Sasaki and Yamada [9]. They found that more regular grid patterns appeared longer. We were pleased when these authors responded to a pre-print of the current article on bioRxiv [25]. They argued that our reflectional symmetry is not equivalent to their grid regularity, and that our failed replication does not reduce confidence their conclusions [26]. We are now agnostic. A new series of experiments may be the only way to determine which non-accidental arrangements reliably elongate subjective duration. Such a research program may confirm the dissociation between reflectional symmetry and grid regularity proposed by Sasaki and Yamada [26].

Null results are often hard to interpret (hence they go unpublished). However, null results are still scientifically relevant, especially when there is no obvious flaw in the study. For instance, null results from an underpowered experiment are less interesting. However, our experiment was not statistically underpowered by conventional standards (a priori power was > 0.8) and there was no hint of a trend in the expected direction. This null result is not likely to be a consequence of low statistical power.

It should be emphasised that what we report here is a failed *conceptual* replication (see Nosek & Errington [27] for careful consideration of different replication types). A failed conceptual replication in experimental psychology could be uninteresting if the stimuli differ from original experiments in some obvious way that, with hindsight, was always very likely to abolish the expected effect. However, there is no obvious stimulus-based explanation for our null result.

Our dot patterns were different from the checkerboards in Palumbo et al. [8] and Ogden et al. [7] in a number of ways. For instance, they were more complex. Complexity can modulate perceived duration [28], so this could be part of the story. However, it is unclear why

increasing complexity should supress the difference between reflection and random conditions. Likewise, our stimuli were 5˚ diameter discs instead of 10˚ squares, but there is no obvious reason why this size change would abolish effect of symmetry on duration either.

Previous studies used durations ranging up to 1.6 seconds, while we used durations <1 second. Could this explain the inconsistencies with previous work? Ogden et al. [7] found the symmetry effect only for durations of less than a second, so it may be necessary to present a range of durations for the effect of symmetry to manifest at the lower end. Indeed, range effects are well known in the timing literature: Vierordt's law states duration estimates tend to cluster round the mean presented (so the shorter durations are overestimated, and longer durations underestimated). Vierordt's law could perhaps contribute to a multifaceted explanation for our null result.

A failed conceptual replication could also be uninteresting if the task had changed in an obviously problematic way. However, we cannot identify an obvious task-based explanation here. Ogden et al. [7] used a verbal estimation task, where participants entered judgements using the keyboard. In contrast, our participants entered judgments using the mouse and continuous scale. Our new response entry method may have reduced 'quantizing' (where participants tend to give round number estimates such as "500 ms" or "1000 ms" [29]). The extra precision could theoretically introduce ceiling effects that mask the effect of symmetry. However, the effect of objective duration on subjective duration was comparable to previous work (Current study $\eta_p^2 = 0.85$, Ogden et al. [7], $\eta_p^2 = 0.88$, Palumbo et al. [8], $\eta_p^2 = 0.76$). This suggests that suppressive ceiling effects were no greater in the current experiment.

The above paragraphs demonstrate how failed conceptual replications can arise from previously unknown boundary conditions. Some change in stimulus or task, originally assumed to be trivial, has the unforeseen consequence of abolishing the expected effect. Palumbo et al. [8] and Ogden et al. [7] might have been lucky by choosing parameters that worked. Indeed, it is common to sell a null result as the positive discovery of a new factor that modulates an established effect. However, it would often require additional experiments to make a convincing case. Another alternative is that the effect was never reliable under any conditions.

In summary, this is failure to replicate cannot be easily explained. We believe such failures should not end up in the file drawer. Timing researchers should not have the mistaken impression that the effect of symmetry on subjective duration has been established without exception.

## Author Contributions

**Conceptualization:** Alexis David James Makin, Marco Bertamini.

**Data curation:** Alexis David James Makin, Afzal Rahman.

**Formal analysis:** Alexis David James Makin.

**Funding acquisition:** Alexis David James Makin, Marco Bertamini.

**Investigation:** Alexis David James Makin.

**Methodology:** Alexis David James Makin, Marco Bertamini.

**Project administration:** Alexis David James Makin, Afzal Rahman.

**Resources:** Alexis David James Makin.

**Software:** Alexis David James Makin.

**Supervision:** Alexis David James Makin.

**Validation:** Alexis David James Makin.

**Visualization:** Alexis David James Makin.

**Writing – original draft:** Alexis David James Makin, Marco Bertamini.

**Writing – review & editing:** Alexis David James Makin.

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
