## [Decision Letter · Decision Letter 0]

11 Nov 2020

PONE-D-20-25416

A failure to replicate the effect of visual symmetry on subjective duration

PLOS ONE

Dear Dr. Makin,

Thank you for submitting your manuscript to PLOS ONE. After careful consideration, we feel that it has merit but does not fully meet PLOS ONE’s publication criteria as it currently stands. Therefore, we invite you to submit a revised version of the manuscript that addresses the points raised during the review process.

We look forward to receiving your revised manuscript.

Kind regards,

Guido Hesselmann

Academic Editor

PLOS ONE

Journal Requirements:

2. PLOS ONE does not copy edit accepted manuscripts. Please proofread for typos and grammar. In addition, please change "female” or "male" to "woman” or "man" as appropriate, when used as a noun."

"This work was partly funded by the Economic and Social Research Council UK award to AM

https://esrc.ukri.org/

ES/S014691/1

Reviewers' comments:

Reviewer's Responses to Questions

**Comments to the Author**

1. Is the manuscript technically sound, and do the data support the conclusions?

Reviewer #1: Partly

Reviewer #2: Yes

2. Has the statistical analysis been performed appropriately and rigorously? 

Reviewer #1: Yes

Reviewer #2: Yes

3. Have the authors made all data underlying the findings in their manuscript fully available?

Reviewer #1: No

Reviewer #2: Yes

4. Is the manuscript presented in an intelligible fashion and written in standard English?

Reviewer #1: Yes

Reviewer #2: Yes

5. Review Comments to the Author

Reviewer #1: Please see uploaded file.

Previous work suggests an effect of visual symmetry on estimates of duration. The present manuscript seeks to extend on these findings by manipulating the number of folds axis reflections) in the stimuli. The justification for using these kinds of stimuli is only briefly provided in terms of “perceptual goodness”. The results suggest that not only is there no evidence for an influence of the new “folds” manipulation, but also the original symmetry bias of duration can’t be reproduced. The paper in general is presented within an metascience argument about the value of presenting nulls to avoid file draw issues.

I think that it’s good that the authors are actively seeking to reduce file drawer issues. I appreciate that they may be concerned that editors and reviewers may not appreciate this and so have presented the paper in a metascience context, largely. I think that more can be done to introduce the motivations that led to the study being run (why did you want to investigate number of folds to begin with?). Providing the materials and data is great. I would appreciate if more effort were made to make the data and analyses (see below) easy to interpret for a reader. There are a few places (both methodologically and conceptually) where I think the paper would benefit from more clarity/details. The one big issue I can see is that claims about being a failed replication or similar are hampered by being a conceptual replication without any evidence of a strong manipulation check. In the discussion the authors start in on addressing this issue, which is great, but only assert that the symmetry in the used stimuli is sufficient. I think that presenting some evidence to support this claim is required. Overall a brief and, other than the above issues, sound paper to my reading.

Reviewer #2: Makin, Rahman, & Bertamini (2020) revisit the question whether durations of symmetrical dot patterns are perceived as longer than those of asymmetrical dot patterns. Dot patterns, either random or containing 1, 2, 3, 4, or 5 symmetrical axes, were presented at varying durations (0.4, 0.6, 0.8 s, in addition to filler trials in which durations ranged from 0.1 to 0.9 s) which were subsequently estimated by 40 observers using a continuous scale from 0 to 1 s. Repeated-measures ANOVAs revealed no effects, neither of symmetry nor number of axes, thus failing to replicate previous effects. The authors argue that this null result is not a consequence of stimulus differences or low statistical power compared to previous studies.

To my mind, the described experiment is technically sound, even though the authors should include further methodological details to facilitate the readers’ understanding. The introduction features a long subsection about the relevance of null effects, yet the description of the relevant studies and their background is short and would – especially given this is a replication – benefit from relevant technical details. The same is valid for the discussion section: A more thorough examination of the differences between the studies in question and the current studies seems necessary, as the interpretation in terms of underlying mechanisms is not exactly obvious. In addition, a clarification on symmetry versus regularity should be included. The conducted analyses provide straightforward and sufficiently strong evidence for the null effect, but (unfortunately) no further attempts were made to scrutinize this result.

Major

(1) The manuscript is written in a way that suggests that previous findings could not be replicated, which may seem like a bold statement given that the studies in question have used (to some extent) different stimuli. Although I agree that to make a diagnostic inference about previous results a replication study need not be exactly similar to previous ones (e.g., Nosek & Errington, 2020), the possible contraints under which an effect might (only) be found, should be discussed. For instance, the most obvious candidate is stimulus complexity: Odgen et al. (2016) used 10x10 grids, as did Palumbo, Odgen et al. (2015a), whereas Palumbo et al. (2015b) used the same stimuli as in the present study, but not in a duration estimation task. Also, virtually all previous studies used longer durations up to 1.6 s, whereas here the longest duration was 0.9 s. This may be important in the context of Vierordt’s Law, suggesting a shorter intervals are more likely to be overestimated if longer reference intervals are present. In addition, could the scale have introduced a central tendency bias (e.g., Douven, 2017) instead of quantizing or ceiling effects? Moreover, what about stimulus size? Previous studies used 10x10 dva² patterns, that is at least twice the size of the stimuli used here (5 dva in diameter). I do not think that these differences invalidate the current study, but they should be both reported and discussed to provide an informative picture to the reader.

(2) Curiously, the manuscript includes the study of Sasaki & Yamada (2017), which uses an stimulus and technique quite different from what was used by the authors’ and their previous studies. In this context, I came across a response by Sasaki & Yamada (2020) to this manuscript, which should be incorporated in a revised version. Like these authors, I share the concern that the underlying mechanisms that give rise to overestimation of duration of symmetrical or regular stimuli, respectively, might be quite different: On the one hand, valence and arousal may play a role, on the other hand, a neural energy account involving second-order orientation processing may contribute to an explanation of the results. Clearly, these two accounts are not mutually exclusive, but they should be discussed thoroughly to avoid misconceptions.

(3) Methodological details should be clarified or added to the manuscript to fulfill the rigor criterion. To just list a few: Were trials presented in a randomly interleaved fashion or blocked? The term N axis is unclear, but I assume it relates to the number of folds. If the SD of the number of dots changes with folds, was that the same for the random stimuli? How were folds included in the random condition? Were the stimulus controlled for luminance, surface area, (or incidental orientations), as in Palumbo, Odgen et al. (2015)? Would there be a possibility to control for these covariates post-hoc, as in a reverse correlation analysis? Furthermore, due to the random nature of filler trials, were their physical durations on average similar? To account for varying durations in these trials, the reported durations could be normalized by their corresponding physical durations, which would also give a more interpretable estimate on the amount of underestimation. The factor physical duration was included in the rmANOVA as a continuous/metric one, do the results change if it is included as a discrete/ordinal one?

Minor

“Therefore, there was no evidence that visual properties of the patterns altered performance” (p. 9): How do population-level correlations between actual and perceived duration indicate whether low-level visual features have an impact on duration estimation? This should be rephrased or elaborated on.

“However, the symmetry was more salient here, and if anything, the effect of symmetry on duration should have been larger” (p. 10): Could the authors elaborate on why this should be the case? First, it is not instantly evident why symmetry should be more salient, given the higher complexity of the stimulus, and, second, is there other evidence that symmetry – varied parametrically – impacts subjective duration specifically?

List of typos or grammatical issues:

- Sasaki & Yamada instead of “Sakaki & Yamada” (p. 5)

- “was more likely abolish the expected effect” (p. 10)

- “For instance null result might be…” (p. 10)

- “well included there were 1-5 folds in the random conditions” (p. 7)

- “Participants sat darkened cubicle” (p. 7)

- “duration estimate in was obtained in all 30 conditions” (p. 8)

6. PLOS authors have the option to publish the peer review history of their article (what does this mean?). If published, this will include your full peer review and any attached files.

Reviewer #1: **Yes: **Warrick Roseboom

Reviewer #2: No

---

## [Author Response · Author response to Decision Letter 0]

25 Nov 2020

Please see attachment for response to reviewers. There are figures are formatting which cannot be embedded in this box.

The changed manuscript and response to reviewers and appear right at the end. The response to reviewers is the very last thing, explaining changes, is right at the end. This is slightly unconventional, but there were restrictions in file upload order and I couldn't reorder the files (at least on my browser).

---

## [Decision Letter · Decision Letter 1]

9 Dec 2020

No effect of multi-axis dot pattern symmetry on subjective duration

PONE-D-20-25416R1

Dear Dr. Makin,

We’re pleased to inform you that your manuscript has been judged scientifically suitable for publication and will be formally accepted for publication once it meets all outstanding technical requirements.

Kind regards,

Guido Hesselmann

Academic Editor

PLOS ONE

Additional Editor Comments (optional):

Both reviewers have raised some minor final comments. Please make sure to address them in your final version of the manuscript or during the proofing process.

Reviewers' comments:

Reviewer's Responses to Questions

**Comments to the Author**

1. If the authors have adequately addressed your comments raised in a previous round of review and you feel that this manuscript is now acceptable for publication, you may indicate that here to bypass the “Comments to the Author” section, enter your conflict of interest statement in the “Confidential to Editor” section, and submit your "Accept" recommendation.

Reviewer #1: (No Response)

Reviewer #2: (No Response)

2. Is the manuscript technically sound, and do the data support the conclusions?

Reviewer #1: Yes

Reviewer #2: Yes

3. Has the statistical analysis been performed appropriately and rigorously? 

Reviewer #1: Yes

Reviewer #2: Yes

4. Have the authors made all data underlying the findings in their manuscript fully available?

Reviewer #1: Yes

Reviewer #2: Yes

5. Is the manuscript presented in an intelligible fashion and written in standard English?

Reviewer #1: Yes

Reviewer #2: Yes

6. Review Comments to the Author

Reviewer #1: Only some minor comments:

Page 4. “Ogden et al. (2016) found that symmetrical patterns were misperceived”

More accurate to say “misreported”?

Page 5. I certainly didn’t intend my previous comments about internal clocks as fishing for a citation. I hope you enjoyed my work if you hadn’t come across it before. I would just say that most people probably wouldn’t agree with the work being characterised as “radically different”. Several people have commented that it is only trivially different from “pacemaker accumulator”. While I would challenge that perception, I think it shows that some consider the alternative perspective as quite trivial.

Page 7. Regarding “power”. Thanks for the added details. For clarity, some people use software packages to calculate power and these sometimes have “output”. I was suggesting that if this were the case here, that the output could be placed online as supporting evidence. This is only about complete transparency for people looking into things so they can clearly follow your reasoning and evidence.

Page 11. “This null result is not simply a consequence of low statistical power.”

More accurate to say “likely not…”?

Page 12. Typo: “However, is unclear”

Sorry I originally missed the location of the raw data. Thanks for making it more obvious.

Warrick Roseboom

Reviewer #2: To my mind, all of my comments were addressed appropriately and changes were included in the manuscript.

Just to briefly respond to "We are not 100% clear what the recommendations are regarding rmANOVA? Our Duration factor had three discrete levels.": Then it would be expected that the factor duration, if treated as a discrete variable with three levels, would have the corresponding degrees of freedom, i.e., DFn=2 and DFd=78, just like the interaction between Regularity and Duration ("(F (2,78)", p. 10) already suggests, or am I missing something here?

7. PLOS authors have the option to publish the peer review history of their article (what does this mean?). If published, this will include your full peer review and any attached files.

Reviewer #1: **Yes: **Warrick Roseboom

Reviewer #2: No

---

## [Editor Report · Acceptance letter]

22 Dec 2020

PONE-D-20-25416R1 

No effect of multi-axis dot pattern symmetry on subjective duration 

Dear Dr. Makin:

I'm pleased to inform you that your manuscript has been deemed suitable for publication in PLOS ONE. Congratulations! Your manuscript is now with our production department. 

Kind regards, 

on behalf of

Dr. Guido Hesselmann 

Academic Editor

PLOS ONE